# Brief communication: Glacier run-off estimation using altimetry derived basin volume change: case study at Humboldt Glacier, Northwest Greenland

Laurence Gray[1]

[1]Department of Geography, Environment and Geomatics, University of Ottawa, Ottawa, K1N 6N5, Canada

*Correspondence to*: Laurence Gray (laurence.gray@uottawa.ca)

**Abstract.** CryoSat can provide temporal height change around the Greenland Ice Sheet including that close to the terminus of many glaciers. Height change from the northern outlet of the Humboldt Glacier in north-western Greenland is combined with ice flux into and out of sections of the glacier basin to derive the water run-off each year from 2011 to 2019. The cumulative nine-year run-off for this part of the Humboldt basin is $9.6 \pm 2.9$ km$^3$ and is predominantly subglacial at the terminus with large run-offs occurring in 2012, 2015 and 2019, and much smaller ones in 2013, 2016, 2017 and 2018.

## 1 Introduction

The recent 21$^{st}$ century increase in mass loss from the Greenland Ice Sheet (Mouginot et al., 2019, The IMBIE team, 2020, Smith et al., 2020) has emphasized the need for regular monitoring of the periphery of the ice sheet, the area which has been, and still is, changing the most rapidly. While glacial ice discharge measurements are now widely available for Greenland outlet glaciers (e.g., King et al., 2018, and Mankoff et al., 2020), direct measurements of the surface and subglacial run-off are not. Water run-off represents an important contribution to the overall mass balance and is normally estimated using a regional climate model, e.g. the MAR model (Fettweis, et al., 2017, 2020).

The interferometric mode on the European Space Agency (ESA) CryoSat satellite was developed in part to alleviate the problems associated with measuring glacial ice height with radar altimeters when surface slopes are relatively large, e.g., at the periphery of the Greenland Ice Sheet. Coherent processing is used in the 'SARIn' mode to achieve an along-track resolution of ~ 380 m, and two cross-track antennas are used for footprint geocoding using interferometric processing (Parrinello et al., 2018). Greenland outlet glacier termini are almost always in a local topographic low such that the 'point-of-closest-approach' (POCA) for a satellite radar altimeter pass across a glacier terminus is often displaced to adjacent higher elevation terrain. This effect coupled with the larger slopes and rough surfaces means that it is difficult to measure the height or height change of glacier termini reliably with the traditional radar altimetry technique; i.e. the estimation of the time-of-arrival of the first radar returns. This suggests that 'swath mode' processing (Gray et al., 2013), which uses the part of the waveform beyond the POCA, may be preferable for estimating the height change in these regions.

## 2 Data and Methods

### 2.1 Height change estimation using CryoSat

CryoSat baseline-C and -D intermediate level SARIn (L1b) data collected from the summer of 2010 to the end of 2019 have been used in this work. Baseline-D files contain small improvements to the satellite roll angle which could affect the output of the swath processing algorithm. However, ESA provided corrections prior to the introduction of baseline-D which were used in the current work. Processing to terrain height and height change was done using the methods described in Gray et al., 2015, 2019. For the lower reaches of the tide-

water glaciers I rely primarily on swath mode processing (Gray et al., 2013, Gourmelen et al., 2019). With swath mode processing it is important to select conditions which will minimize the contribution from any unwanted range ambiguous region. For example, when the height and height change of the surface of supra-glacial lakes were mapped using CryoSat swath mode data (Gray et al., 2017), the relatively flat lake provided a strong reflecting surface so that the returns from the lake dominated over any range-ambiguous regions. Consequently, the differential phase reflected the cross-track look angle for the supra-glacial lake and allowed accurate geocoding and height estimation of the lake. In summer, there is a strong probability of wet ice conditions close to the termini of tidewater glaciers, and this can lead to strong specular reflections. By selecting relatively strong waveform returns (> -150 dB) from summer passes there is a strong likelihood that the wet glacier surface will dominate the composite return signal and the differential phase will then more accurately reflect the cross-track look angle and allow accurate geo-coding.

In calculating ice height and temporal height change we need to be able to change both the area over which the change will be measured and also the time interval between average height estimates. There is a trade-off between the size of the patch over which the SARIn data are binned and the ability to extract temporal variation in height change. The shortest practical temporal sampling possible with CryoSat is the 30-day sub-cycle but then a relatively large area is preferred to get adequate averaging (Gray et al., 2019). That approach is not appropriate for the relatively small areas close to the terminus of the Humboldt Glacier so the size of the temporal window has been increased to obtain more samples for averaging. Here, the swath mode height data are sorted into overlapping spatial windows of size 2.4 km by 2.4 km sampled at 1.2 km bin spacing, and a search is done for closely spaced height estimates (< 100 m apart) in a succession of time periods. For the year-to-year height change, data from July 1 to October 15 were used to capture the possibility of strong returns and to minimize the possibility of a varying bias between the surface and the CryoSat detected height (Gray et al., 2019). While the surface height at any point on the glacier will likely change during this period, the relatively large temporal bin is not as big a problem as it might appear. The CryoSat satellite orbit repeats every 369 days, so that when comparing the data from one year to the next, many of the pairs will be close to one year apart in time, and the average of the height change of all the possible pairs within the separation criteria will reflect the yearly height change even when using a temporal window of three and a half months each year. The unit of time for the year-to-year differences is then early fall in one year to the same period in the next.

After binning the height estimates into the 30-day or the year-to-year time periods, the average height change between all the possible time periods is obtained. For any two time periods the average height change is calculated from the height differences of the closely spaced points. The height change between any two periods is then calculated from the matrix of average height changes using the method outlined in Gray et al. 2015. The result is the weighted average of the direct comparison of two time periods combined with the differences using a third time period. For example, the height change between year 1 and 2 for a specific area uses the height difference for those two years but also the height changes from year 1 and year 3, and year 2 and year 3. Because the data points used in the various comparisons are different, this process gives an independent estimate and leads to a better overall average. In the current work the height difference between any two years in the 10-year span between the Falls of 2010 and 2019 uses a weighted average of the direct difference and the eight differences using a third period. Further details are available from Gray et al. 2015.

While swath mode processing provides results close to the glacier terminus in the region in which the POCA algorithms struggle, at higher elevations the surfaces are smoother, the slopes more moderate, and POCA results tend to have lower random errors and a lower probability of bias errors (Gray et al., 2017). For this reason, POCA results were used at elevations above ~600 m. The method used is similar to the swath processing approach outlined above, although now the spatial bin size is doubled to 4.8 km, the bin spacing is 2.4 km, and the search for pairs of points in the different time bins is carried out for separations less than 400 m. The GIMP DEM (Howat et al., 2014) is used for slope correction (Gray et al., 2019).

**2.2 Mass change estimation using CryoSat**

Satellite altimetry can provide volume change data directly, but not mass change. Often models are used to estimate the near surface density change and surface lowering in the accumulation zone associated with different summer conditions (Reeh et al., 2005, McMillan et al., 2016), this can then allow the estimation of mass change. However, any model will depend on accurate weather data which is in limited supply around Greenland, and this applies to the Humboldt Glacier basin. Here I use the CryoSat data itself to estimate the surface lowering in the summer months in those areas in the accumulation zone where there is no evidence of direct run-off through surface streams or moulins, and no evidence of a change in ice flux which could explain the height loss. Using the MAR model for 2010 to 2019, the elevation of the transition between net ablation and net accumulation varies in this area from ~ 1000 m to ~ 1300 m. Consequently, it is necessary to consider the effect of firn densification only at the higher elevations in our study area. To search for height change associated with firn densification it is necessary to use the 30-day temporal sampling and a relatively large area in the accumulation zone, $> \sim 10^3$ km$^2$. Using a relatively large area is not an issue as conditions in the accumulation zone vary relatively slowly with position. Further, there are now 50 m resolution SAR data sets (Joughin, 2019) with repeat coverage every 6 days which show the initiation, extent, and termination of the summer melt. Also, this imagery (see accompanying material) shows the positions of supra-glacial lakes and surface run-off streams. Consequently, when there is no evidence of extensive surface run-off to a lower elevation or change in surface ice speed it is possible to associate a relatively fast summer height loss over a large area in the accumulation zone with firn densification. This provides a straightforward method of correcting the volume loss to obtain an estimate of mass loss.

**2.3 Ice velocity and gate flux estimation**

Ice velocity data were obtained primarily from the NSIDC MEaSUREs (NASA's Making Earth System data records for Use in Research Environments) web site. Data from both radar (NSIDC-0481, -0478 and -0731, Joughin et al., 2018, 2020) and optical satellites (NSIDC-0646, Howat, 2017) have been used. For 2019, ice velocity data sets from the Programme for Monitoring of the Greenland Ice Sheet (PROMICE) have also been used (Solgaard and Kusk, 2019). Although the relative accuracy of the velocities derived from the optical satellites is lower than that from speckle tracking with SAR data, they are necessary to capture the speed changes in the summer melt season. The 'InSAR Selected Glacier Site Velocity Maps' produced from image pairs from the German Aerospace Centre's (DLR) twin satellites TerraSAR-X / TanDEM-X (TSX, Joughin et al., 2020) tend to have the lowest error estimates. Plots of the TSX speed variation across the terminus gate (gate 1 in Fig. 1) show that the shape of the velocity profile does not change with year or season although the magnitude does. For this reason, a reference point was selected and the speed across the gate from all the various sources was based on the reference TSX velocity profile times the ratio of the velocities at the reference point. The temporal frequency of velocity estimates increases after 2014 and the spike in summer surface velocity can be readily tracked. However, prior to 2014 we know there was a spike in the summer gate velocity, but the timing and duration were not available from satellite imagery. GPS receivers had been used to track the seasonal change in surface speeds of eight Greenland outlet glaciers (Ahlstrøm et al., 2013), including one close to the terminus of the Humboldt Glacier. Unfortunately, this receiver was lost at the beginning of July 2012, but the onset and peak of the velocity spike were captured (Fig 9, Ahlstrøm et al., 2013). With this as a guide some extra velocity points were added for the summers of 2011 and 2012 to limit the temporal duration of the velocity peak measured by the optical satellites. The temporal velocity data at the reference point were up sampled using linear interpolation to give daily speed estimates.

The gate close to the terminus (Gate 1 in Figure 1) is ~10.2 km wide with an average ice thickness of 360 m in 2011 decreasing to 321 m in 2019. The net ice flux through gate 1 is calculated by summing 75 flux estimates each using the average speed and ice thickness over a segment of width 137 m. The average ice thicknesses for gates 2, 3 and 4 are ~ 926, 1157 and 1422 m, the average speeds are ~ 94, 56 and 35 m/year, and the cross-gate flux calculations are performed for segments of width 170, 187 and 2000 m. The relatively wide segment for gate 4 was possible because of the nearly linear velocity variation across the gate. The flow lines were derived from

the x- and y- components of the 200 m 2017-2018 velocity mosaic of Greenland (Joughin et al., 2015). The upstream end of the test area (gate 4 in Fig. 1) is orthogonal to the flow direction, and the flow lines are derived from points on this gate separated by 2 km. When calculating the flux, the surface speed is normally used as the depth averaged speed whenever the surface speed is ~ 100 m/year or larger (Mankoff et al., 2020). The flux estimates through gates 3 and 4 have used fractions of the surface speed to account for the possibility that the depth averaged speed is less than the surface speed. For gate 3 the fractions were 0.95, .975, and 1, and 0.9, .95 and 1 for gate 4. The different fractions are used to estimate the potential error arising from the uncertainty in the depth averaged velocity. 'BedMachine' data (Morlighem et al., 2017) were used primarily for the ice thickness data. However, the estimated ice thickness uncertainty was quite large for part of the terminus gate and interpolated IceBridge data were used instead. The 20 April 2013 IceBridge flight provided 10 lines separated by ~ 2 km in the flow direction over the northern Humboldt Glacier terminus and this allowed revised ice thickness using the MCoRDS data (Paden et al., 2019). The cross-sectional area of the other gates and associated errors were based on the BedMachine data. Finally, the daily ice flux through the various gates can be summed to give nine yearly values from the fall of 2010 to the fall of 2019.

## 2.4 Water run-off estimation

The water run-off is estimated based on mass conservation. Surface height change data can be used to estimate the volume and mass change for that part of an outlet glacier basin defined by input and output crossflow gates connected by ice flow lines. The input expressed as mass per unit time is the sum of the ice, firn, and water flux at an up-stream gate plus the contribution from surface precipitation, i.e.,

$$M_{in} = \rho_i F_{ice\_in} + \rho_w F_{water\_in} + \rho_f F_{firn\_in} + A\rho_w acc, \tag{1}$$

where $F_{ice\_in}$, $F_{firn\_in}$ and $F_{water\_in}$ are the yearly input fluxes of glacial ice, firn and water respectively. $A$ is the surface area, $acc$ is the yearly accumulation in water equivalent height, $\rho_i$, $\rho_w$, and $\rho_f$ are the average densities of the ice, water, and firn components and the unit of time is Sept. 1 in one year to Aug. 30 in the next. The yearly ice flux at a down-stream gate close to the glacier terminus can be estimated assuming that the gate cross-section is occupied by glacial ice and the possible fraction of air, snow or water in the gate cross-section is small enough in relation to the potential error in the gate cross-section that they can be ignored.

The mass change, or mass balance for a specific area, can be equated to the difference of input and output masses, $\delta M = M_{in} - M_{out}$, where $\delta M$ is estimated based on changing surface heights derived from the CryoSat data. Here the area of the large basin extends up into the percolation zone so that while the height being measured is essentially the glacier surface, the density of the upper layers may change depending on the history of surface melt. The appropriate correction to the mass balance is based on the analysis of the CryoSat data, as outlined above in section 2.1, and described in more detail in the results for the northern arm of the Humboldt Glacier below. The total output at the terminus gate is $M_{out} = \rho_i F_{ice\_out} + \rho_w F_{water\_out}$ so the excess water run-off for the defined part of the glacier basin can be calculated as

$$\rho_w(F_{water\ out} - F_{water\ in}) = \rho_i(F_{ice\ in} - F_{ice\ out}) + A\rho_w acc - \delta M. \tag{2}$$

If the ice is much thicker at the upstream gates, as is the case with the Humboldt Glacier, the contribution of firn to the input flux is relatively small and much smaller than the potential error introduced by the uncertainties in ice thickness and the depth averaged velocity. The mass loss from evaporation and sublimation are also neglected as they are relatively small, and again, much less than the errors in the estimates of mass change and the difference in flux estimates.

## 2.5 Accumulation estimation

The MARv3.11 regional climate model of Greenland (Fettweis, 2020) has been used to provide the estimates of the yearly accumulation over the studied area of the Humboldt basin. These data are provided daily, sampled at 20 km and are provided in units of  mm. water

equivalent. Here the data are up sampled to the BedMachine grid (provided in polar stereographic coordinates on a 150 m grid, Morlighem et al., 2017) using the Matlab function ScatteredInterpolant. Yearly accumulation data from September 1 to August 30 are summed for the three 'sub-basins' (Fig. 1) beginning with the 2010-2011 year and ending with 2018-2019.

## 3 Results

### 3.1 Run-off for the northern arm of the Humboldt Glacier

The calving front of the northern arm of the Humboldt Glacier has receded since 1975 (Carr et al., 2015) and, in common with many of the outlet glaciers on the west and north coasts of Greenland, the speed close to the terminus has also increased since ~ 2000 (Joughin et al., 2017, Hill et al., 2018). Figure 1 shows the position of the study area in the Humboldt Glacier basin and in Greenland. The four gates and the outer flow lines define 3 basins, the largest basin, no. 3, is contained within the outer flow lines and gates 1 and 4, while gates 2 and 3 define the upstream ends of the smaller basins, 1 and 2 respectively. The nine-year height loss, Fall 2010 to Fall 2019, is illustrated as a colour overlay on a SAR image (Sentinel 1 from 27 July 2019). Note that the colour bar is non-linear and the bulk of the height loss, up to ~ 45m, is close to the terminus. During this period, the net mass loss from basin 3 was 11.3 ± 0.6 Gt but more than 50% of this was lost within 20 km of the glacier terminus.

The velocity data plotted in Fig. 2A show the speed at the reference point in gate 1 (Fig. 1) and the quite dramatic increase in summer velocity occurring after the onset of the melt period. From a sequence of 2019 Sentinel images in this area (see supporting material) the onset of melt in 2019 began after June 6 but before June 13, and the image from June 13 shows indications of wet snow up to an elevation of ~ 1300m. The first significant speed increase is plotted here with a nominal date of Jun 17 but this was derived from passes on June 5 and June 29 so we cannot be certain when in this period the relatively sudden jump in speed occurred. However, from Fig. 9, Ahlstrøm et al., 2013, we know that the 2012 speed increase happened over a few days at the beginning of July. Some additional velocity values have been added to Fig. 2A to constrain the period of the speed-up so that it is consistent with the later years when there were more temporal speed data. Although there are fewer velocity data available for the upstream gates 2, 3 and 4 both the seasonal and year-to-year variations are very much smaller than those exhibited at the gate close to the terminus.

In calculating the ice fluxes through the four gates, the reduction in ice thickness over the nine years was accounted for although the larger ice thickness and much smaller thinning rates at gates 3 and 4 lead to a small year-to-year correction. The cumulative ice flux at the terminus gate 1 is actually less than that estimated through the other three gates (Fig. 2B) up to ~2016, but due to the increase in average speed beginning after the 2014 summer, the cumulative ice flux across gate 1 exceeds that from any of the other gates by the summer of 2018. The ice flux through gate 1 between Fall 2018 and Fall 2019 was 3.5 ± 0.2 $km^3$, more than twice the flux between fall 2010 and 2011. The ice volume loss in $km^3$ and the net mass balance in Gt for the three basins are shown in Figure 2C. A density of 910 $kg/m^3$ was used in the conversion of volume to mass. All the years exhibit a negative mass balance, except the fall 2012 to fall 2013 year which showed a small positive mass balance. Mass loss was largest for 2019 due to the unusual Greenland weather (Tedesco and Fettweis, 2020), and to the increased speed at the terminus.

The cumulative net run-off for the three basins (Fig. 2D) is estimated based on the ice flux difference between input and output, the accumulation and the net change in basin mass, as described in section 2.4 above. By the Fall of 2019, the cumulative run-off for basin 2 is comparable to that for the larger basin 3. As the larger basin contains the smaller one, the run-off from the larger basin cannot be less than the smaller one implying that most of the run-off originates from below gate 3 in all years. However, when converting the yearly volume change to mass change in the accumulation zone care should be taken to account for changing summer weather conditions and the impact this may have on firn compaction and therefore, near surface density.

Firn densification models can be used to improve the volume to mass change estimation, e.g. McMillan, et al. (2016) and Smith et al., (2020), but the results can only be as good as the reanalysis of the input weather data, which are very sparce for the large ice sheets. Here, a straightforward correction has been carried for three years when anomalous height decreases were observed for the summers of 2012, 2015 and 2019. Figure 3 shows the positions of 44,756 height estimates above 1300 m in basin 3, and Fig. 3B shows the average height change sampled at 30-day intervals from the Fall of 2010 to the Fall of 2019. The three red arrows indicate the anomalous height decreases in the summers of 2012, 2015 and 2019. In an idealized situation, the surface height would not change for an ice sheet in equilibrium, and the slow snow accumulation would be balanced by the slow downslope movement of the ice. However, the detected height change data can be affected by temporal changes in accumulation, downslope ice speed and near surface conditions including summer firn densification. A sequence of Sentinel SAR imagery spanning the summer of 2019 (see the supporting material) shows that there was surface snow melt extending up into the accumulation zone of the test area in this year. The NSIDC 'Greenland Ice Sheet Today' web site documents the melt conditions over Greenland and the unusual conditions in this area in the summers of 2012, 2015 and 2019. The unusually warm conditions for 2012 are well known. For the summer 2015; (from http://nsidc.org/greenland-today/2015/11/), '….a surge in melting in late June and all of July as very warm conditions prevailed along the far northern and north-western coast,…'. And for 2019, from http://nsidc.org/greenland-today/2019/11/ ; '…(melting) was particularly intense along the northern edge of the ice sheet, where compared to the 1981 to 2010 average, melting occurred for an additional 35 days. Consequently, the anomalous height decreases in this area can be linked to the unusually warm summers in 2012, 2015 and 2019. These decreases are much larger than would be expected due to the normal reduced accumulation during the summer. While the height decreases could be due to a relatively sudden change in downstream ice speed no such summer spike in speed has been observed at these elevations. As there are none of the clues that one would normally associate with run-off to a lower elevation, e.g., surface streams or supra-glacial lakes, the most likely explanation for these three summer height decreases is surface melting, percolation of the melt water and subsurface refreezing. When calculating the volume change to mass change, I assume therefore, that the height losses of 0.42 ±0.08 m (2012), 0.45 ±0.08 m (2015) and 0.4 ±0.08 m (2019) were due primarily to firn densification. Essentially, this volume loss is due to a loss of air, not ice, so no mass loss is associated with this change. The error associated with this assumption is hard to evaluate but an additional error of ±10 cm has been included to account for unknown biases in the height data (section 3.2 below).

Figure 2E shows the run-off at gate 1 for each year from both basins 1 and 2. Basin 2 run-off peaked in 2012 and 2019 with values approaching 2 km$^3$, 2011 and 2015 also had relatively large values ~ 1.3 – 1.5 km$^3$, and the lowest run-offs (~ 0.4 – 0.6 km$^3$) occurred in 2013, 2016, 2017, and 2018. Although the uncertainty estimates in Fig. 2E are relatively large, it is still gratifying to see that in the years with low run-offs the difference between the basin 1 and 2 run-offs also decreases. Indeed, the small difference implies that in the four years with low run-off it originated primarily from basin 1.

**3.2 Errors**

Errors can be introduced into the estimates for excess run-off through errors in the four terms in equation (2); the input and output ice fluxes, the accumulation, and the mass change. These are derived from ice velocity, ice thickness, the integrated accumulation over the various basins, and the volume to mass change derived from the CryoSat heights. The error in mass change includes the potential error in converting the volume change to mass change associated with variable near surface firn densification. If we assume that the errors in the four terms are independent, then the error in the run-off can be estimated as the square root of the sum of the squares of the error in the four individual terms. The MEaSUREs velocity, PROMICE velocity and BedMachine data have associated error estimates and these data have been used to estimate the errors in the fluxes through the four gates. Normally, the error in gate flux is dominated by the error in the ice thickness (Mankoff et al., 2020). However, for gates 3 and 4 the surface speeds are relatively low, ~57 m/year and ~35 m/year, and it is possible that the depth averaged velocity is a fraction of the surface velocity. By choosing a range for the possible fraction, 0.9,

.95 and 1 for gate 4 and 0.95, .975 and 1 for gate 3, the possible impact of the fraction variation can be evaluated. The uncertainty in the depth averaged velocity for gates 3 and 4 then becomes the dominant source of error for the ice flux through these gates.

The potential error in the volume change term in the equation for run-off arises from errors in the CryoSat heights. For this data, the average standard error of the mean of the POCA and swath-mode heights were ~8 and ~20 cm, respectively. However, this leads to an optimistic picture for the precision of the height measurement as it ignores the possibility of varying bias errors creeping into the results (Gray et al., 2019). Using summer-fall data should minimize the error due to a changing bias between the actual surface and the height derived by the processing algorithm (Gray et al., 2019). Nevertheless, an additional possible bias error of ±10 cm was added to the standard error in calculating the overall error in the yearly water volume changes. The ±20% error in the estimate of net accumulation over the 3 sub-basins is based on the comparison of accumulation made by the IceBridge snow radar and the MAR regional climate model (Koenig et al., 2016).

It is important to note the difference in the way that errors propagate for the different components of the equation for run-off. The mass change term is based on the altimetry derived volume change and the error in the comparing the volume between any two years is approximately the same, i.e., the error in the volume or mass change between 2011 and 2012 is essentially the same as between 2011 and 2019. But this is not the case for the flux estimates where an error in e.g., the cross-sectional area of an ice gate, will accumulate with time such that the error in flux over nine years could be nine times the error over one year. This is the source of the increasing error estimates with time in Fig. 2D.

## 4 Discussion

### 4.1 Velocity change at the terminus over the nine years

Each year, the velocity data show a spike in summer speed at the terminus gate associated with the melt water production. The sudden speed up, and the apparent lack of surface run-off channels near the terminus gate suggest that here the bulk of the run-off happens sub-glacially. The summer variation in surface velocity appears to fit the pattern of speed up associated with increasing basal water pressure in a distributed, inefficient subglacial drainage system, followed by a transition to a more efficient drainage system as channels develop. With the improved drainage system, the sub-glacial water run-off increases but the surface speed decreases as the basal water pressure falls and basal drag increases (Flowers, 2016). As well as the yearly spike in surface speed near the terminus, there is a steady increase in speed after the 2013 melt period which continues through all the seasons until after the 2019 melt when the speed reverts to a value less than the value prior to the 2019 spike in velocity. The cause of the year-to-year speed increase at gate 1 after the 2013 summer speed spike may be related to a reduction in basal drag associated with steady thinning in the terminus region, nominally around 4 - 5 m/year. The question then is why there has been such a large loss of ice through both retreat and thinning at the glacier terminus in such a short time. As a tide-water glacier this involves ice-ocean interactions (Rignot et al., 2016, Flowers, 2018), ice-sea ice interactions (Joughin et al. 2020) and the detailed basal topography (Carr et al., 2015). These issues are beyond the scope of this paper.

### 4.2 Water run-off: source and timing

Measuring basin run-off in this way depends on the difference in ice flux at two gates, one upstream but connected by flow lines to the lower gate near the terminus. If the cross-section at either gate is occupied by water and if that changes year-to-year then there is a possibility of a bias error in the run-off estimate. By picking the Fall time period for the year-to-year volume and flux estimates, it is reasonable to assume that any subglacial water flow that still exists is relatively efficient and that the fraction of the gate cross-section occupied by the water would be small. Certainly, the result that the estimated run-off peaked in years with large surface melts suggests an efficient subglacial hydrologic system year-to-year. Further, the short spike in speed each summer suggests a relatively rapid

transition to the more efficient channelized subglacial outflow. Consequently, even if a fraction of the cross-section of gate 1 in the Fall is not solid ice, the impact on this method should be small.

## 4.3 Applicability of this method

Although data on ice thickness, surface height and velocity are now readily available on the internet, this method for the estimation of water run-off associated with a large glacier does have some limitations; the downstream gate close to the glacier terminus needs to be relatively large (~ 10+ km), and ice thickness and velocity data are required for all the gates. While ice thickness close to the terminus has been measured for many of the large Greenland outlet glaciers, often the ice thickness upstream has not been estimated as accurately. As described earlier, obtaining height change data close to the terminus of many of Greenland's larger glaciers can be challenging due to the relatively large slopes and surface roughness. The derivation of the height change at the gates improves with the number of height estimates. Consequently, the error associated with gate height change may increase in southern Greenland as the coverage by CryoSat degrades due to the sub-satellite track divergence.

 If there was a steady input of englacial or subglacial water at the upper gate and an equal amount of water leaving at the lower gate at the same time this would not be detected with this method. However, around Greenland there is both ice discharge and water run-off, and, as the run-off is predominantly seasonal, it can be estimated with this method. However, the method will be more challenging for those areas and glaciers with strongly divergent upstream flowlines.

## 5 Conclusions

In this paper I show that the interferometric mode of the CryoSat radar altimeter can be used to measure the change in volume of part of the basin of a relatively fast flowing Greenland glacier on a yearly basis. By combining this with ice flux measurements, yearly accumulation estimates and a correction for firn densification related to unusual melt into the accumulation zone, it is possible for the first time to derive useful estimates of the yearly water run-off directly from satellite data.

This approach also permits a direct comparison of water run-off and mass discharge as icebergs. Initially, the run-off in this basin was comparable to the ice discharge, however, after the increase in gate 1 speed in the fall of 2014 the ice discharge exceeded the run-off such that by the fall of 2019 the cumulative ice discharge over the nine years was about two times larger than the run-off. Not surprisingly, most of the run-off originated from the ablation zone. This reduces the potential errors in run-off which could arise from a bias in mass balance estimate due to problems with the surface densification in the percolation zone and also any bias error due to errors in the depth-averaged ice velocity at the upper gate.

There is a relatively large variation in run-off year-to-year, e.g., the run-off in 2012 was about four times larger than that in 2013, although the ice flux was comparable in the two years. This highlights the benefit of a methodology which provides results on a yearly basis and allows comparison with year-to-year conditions. This is preferable to the approach which averages the mass balance over many years, or uses a model for the seasonal or year-to-year variation.

The new generation of satellite altimeters, CryoSat launched in 2010 and now IceSAT-2 launched in 2018, provide height and height change data which allow mass balance estimation at better spatial and temporal resolutions than was possible prior to the CryoSat launch. The use of swath-mode processing for the low elevations at the terminus of the Humboldt Glacier helped provide the height change data necessary to capture the large height loss in this region. Finally, 30-day temporal height change data from the accumulation zone, coupled with other weather information, were used to suggest a link between anomalous height loss in unusually warm summers with firn densification.

Satellite radars and laser altimeters can track temporal height change associated with the sometimes-episodic movement of subglacial water. It is conceivable that, over time, the new altimeter systems and this approach could also be used to estimate the source, movement and volume of subglacial water. In the current example, it is clear that significant flux of water only begins downstream of gate 3 and that in these nine years more than 70% originated below gate 2. In low melt years virtually all the run-off originated from below gate 2.

**Acknowledgements**

This work was supported by the European Space Agency through the provision of CryoSat and Copernicus Sentinel-1 SAR data. The velocity data was provided primarily by the NASA MEaSUREs programme and available from the NSIDC web site. NASA supported the IceBridge (OIB) flights over Greenland, while NSIDC facilitated provision of the BedMachine and airborne OIB data. Ice velocity data for 2019 were produced as part of the Programme for Monitoring of the Greenland Ice Sheet (PROMICE) using Copernicus Sentinel-1 SAR images distributed by ESA and were provided by the Geological Survey of Denmark and Greenland (GEUS) at http://www.promice.dk. NSIDC is acknowledged for the web site 'Greenland Ice Sheet Today' produced by Ted Scambos, Julienne Stroeve, and Lora Koenig with support from NASA and the collaboration of Jason Box, Xavier Fettweis, and Thomas Mote. Comments provided by Dave Burgess, GSC Canada, two anonymous reviewers and the editor, Dr Matsuoka, are gratefully acknowledged.

**Data Availability**

**The various MEaSUREs velocity data sets are available from NSIDC (https://nsidc.org/data/measures/data_summaries), the IceBridge ice thickness data at https://doi.org/10.5067/GDQ0CUCVTE2Q (Paden et al., 2010). CryoSat data are available from the European Space Agency (https://science-pds.cryosat.esa.int/, last access: 31 April 2019; ESA, 2019). BedMachine data (Version 3) are available from https://nsidc.org/data/IDBMG4. The MAR data for Greenland can be download from ftp://ftp.climato.be/fettweis/MARv3.10/Greenland/NCEP1_1948-2019_20km/ The temporal sequence of 2019 Sentinel images are available from the supporting material. Other data and code are available from the author upon request.**

**Author Contribution**

L. Gray completed all the work for this publication.

**Competing Interests**

There are no competing interests involved in publication of this work.

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

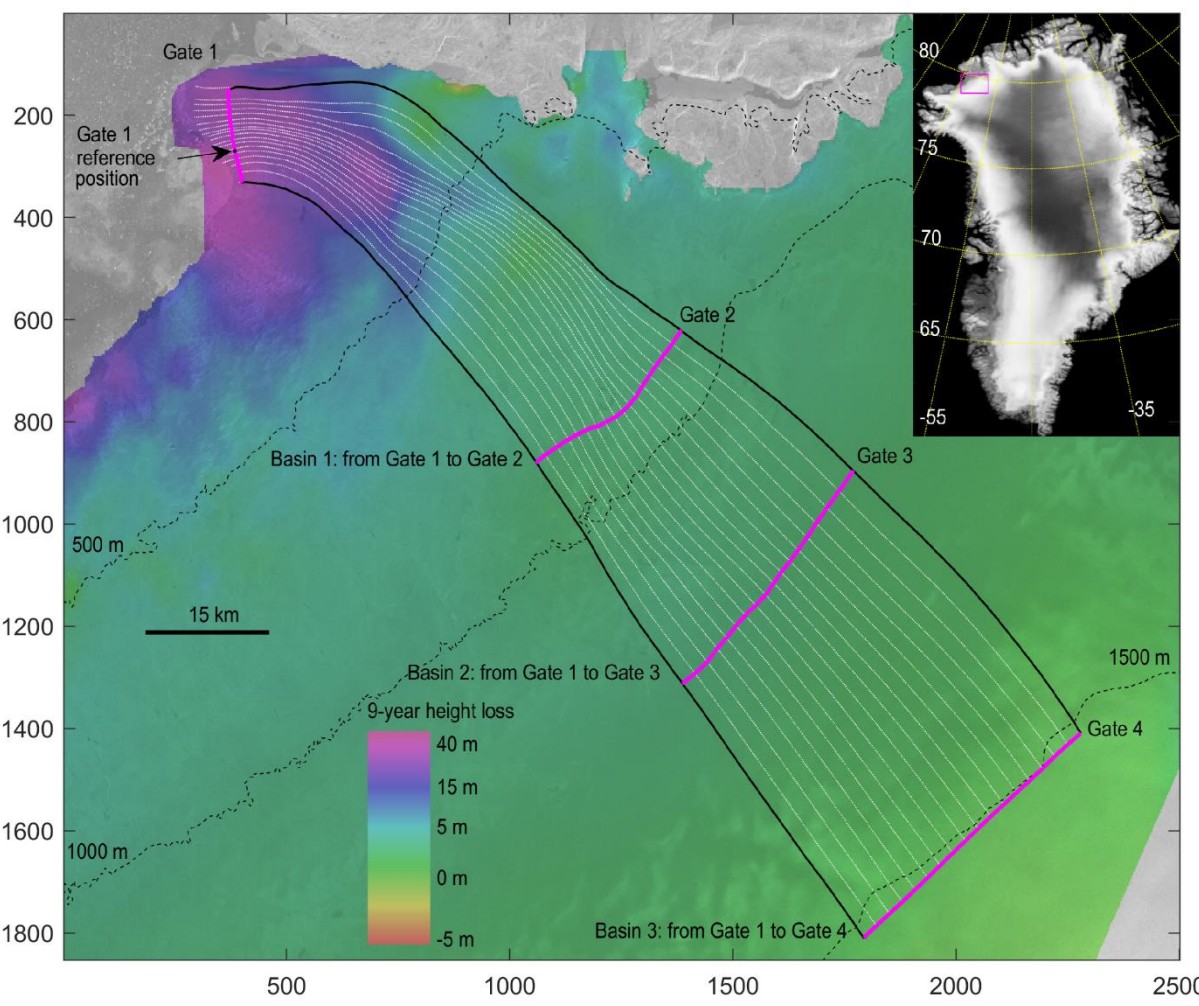

**Figure 1: The positions of the 3 basins are defined by the 4 gates and the outer black flow lines. The white lines are also flow lines derived from the x- and y-components of the 2018-2019 200m MEaSUREs velocity product. The 9-year height loss is illustrated by the colour overlay on the Sentinel SAR image from 27 July 2019. The position of the test area is shown in the insert image by the magenta box in the NW corner of Greenland.**

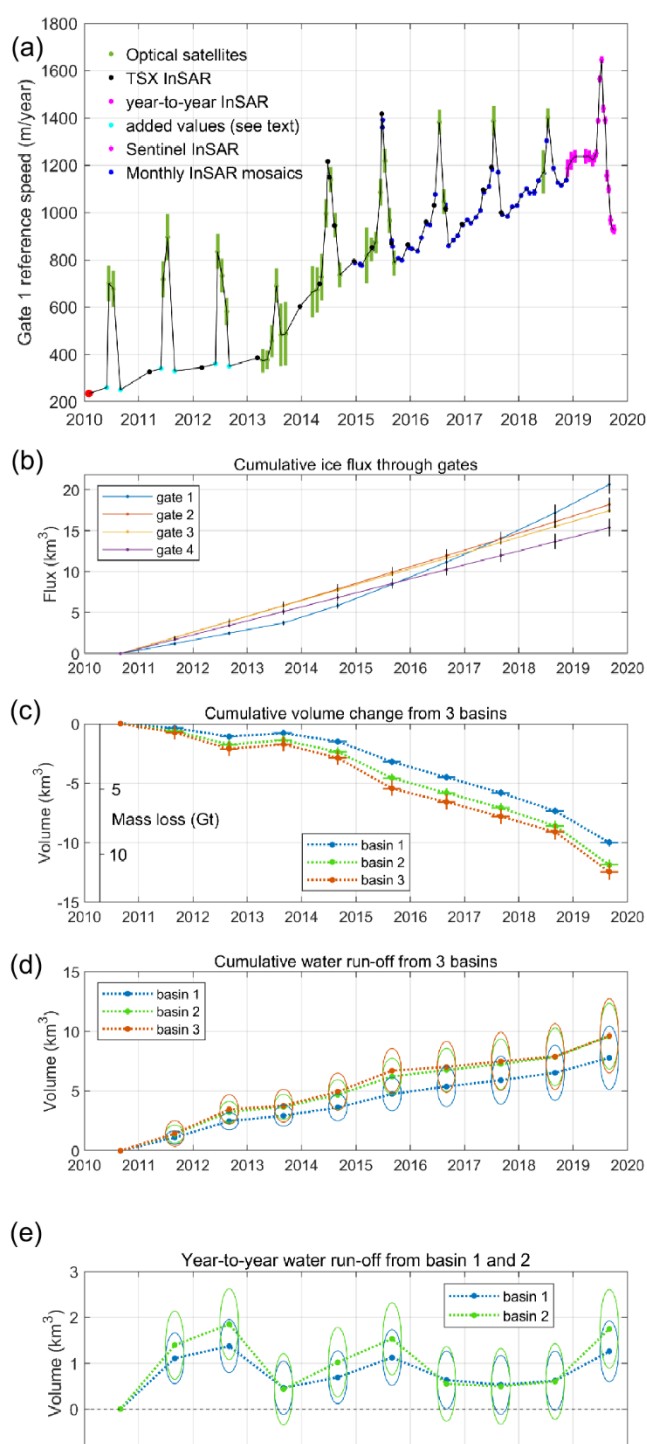

**Figure 2: A: Temporal variation in surface speed at the reference point on gate 1. B: Cumulative year-to-year ice flux through the 4 gates. C: Cumulative volume of ice and mass lost by the three basins over the nine years. D: Cumulative water run-off from the three basins. The ellipses around each point are an indication of (vertically) the potentially uncertainty in the result and (horizontally) the time period of the CryoSat data used in estimating the volume change. E: Year-to-year variation in water run-off from basins 1 and 2.**

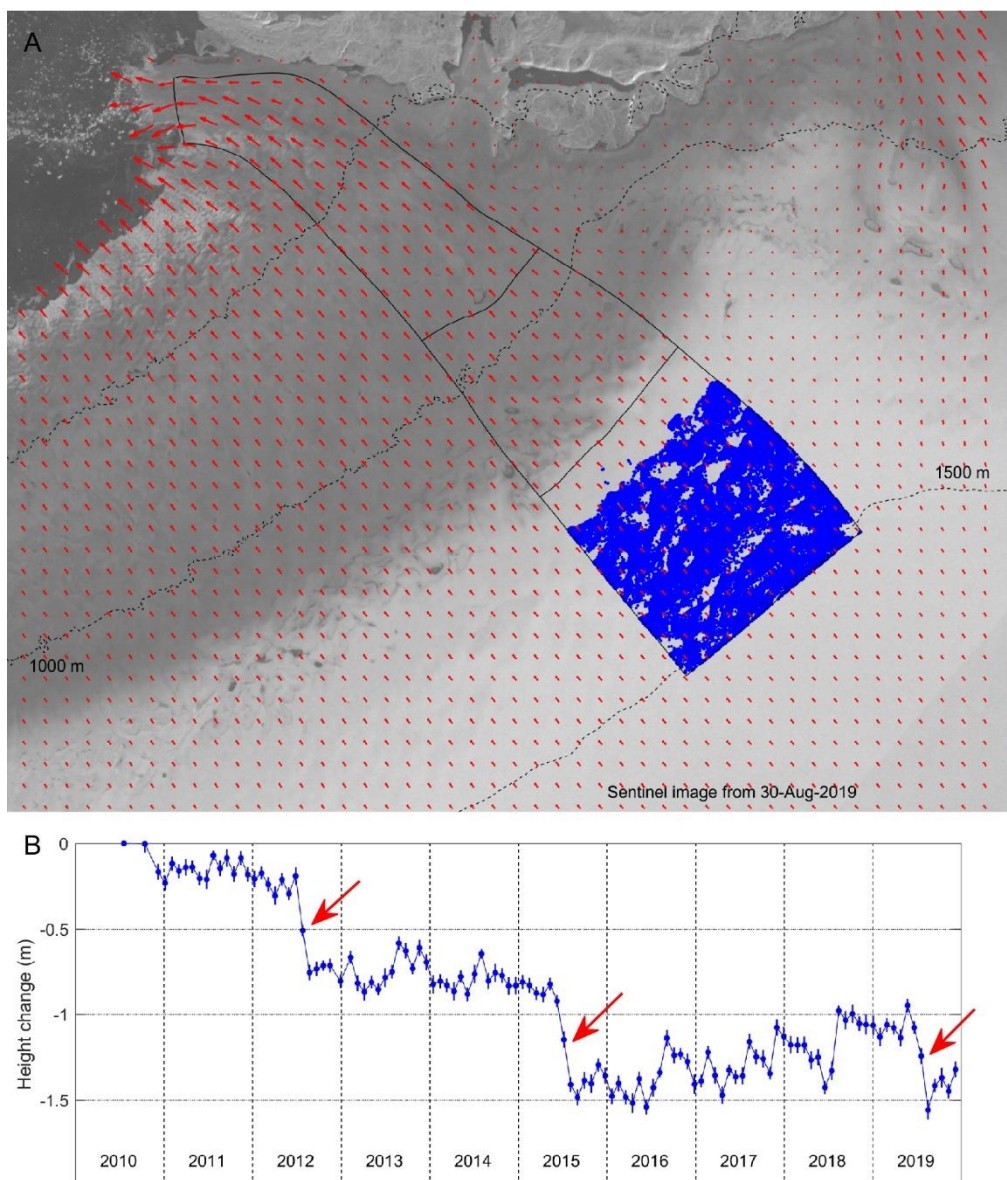

Figure 3. A: Part of the 30-Aug. 2019 Sentinel SAR image showing the positions (blue dots) of the 44,756 CryoSat POCA height estimates which were used in deriving the average 30-day temporal height change plot shown in B. The short vertical lines at each point are ±2 times
the standard error of the mean of the height change estimates for each point and are an indication of the random error in the results. The three red arrows highlight the anomalous average summer height loss in 2012, 2015 and 2019 that has been ascribed to firn compaction due to warm temperatures at these elevations in these years.