# Peer review of "Brief communication: Glacier run-off estimation using altimetry derived basin volume change: case study at Humboldt Glacier, Northwest Greenland"

_The Cryosphere, 2020_

## Referee Comment (RC1) · Anonymous Referee #1 · 23 Oct 2020

This paper uses CryoSat-2 radar altimetry (and OIB ice thickness data and MEaSUREs velocity data) to determine glacier runoff in a northern Greenland glacier basin. The Humboldt is used as a test case for this method.

One general comment that I have is that there needs to be more of a discussion on where this method may be applicable. The author describes the melt/water environment as being the key limitation. I'd suggest that here are other limitations that include: latitude, trend and width of the glacier valley, and OIB data availability.

While this is intended to be a brief communication, the Height change estimation using CryoSat section probably needs more information to have this paper stand a little more on its own.

The paper is fairly well written, although it needs a bit of polish that usually comes from having multiple authors and multiple sets of eyes on a manuscript.

I don't think that the title should have a period

L19: "… synthetic aperture radar (SAR), interferometric SAR (SARIn)…"

L30: I am concerned about differences between baseline C and D. Did you see this? How would differences affect these results?

L36-38: This sentence is a bit long and hence a bit confusing. Needs an edit.

L43-50: Somewhere in here, it might be good to acknowledge that this outlet glacier is at a high latitude (79.5 N), providing more data points than you would get from the southern part of the island, simply based on converging satellite ground tracks. It might be good to add a grid or some indication of this (latitude/longitude) in your figure(s). Further, this is a large glacier, with a wide terminus, providing more data points than you would get from more constrained outlets.

L49: Why was this time range selected? It seems a little later (both start and finish) than I expected. I now see (on L242) why the October date is chosen; what about the start?

L60: Where is the 600 m contour? Below 1000 m the contours can bunch up quickly. This contour (or perhaps the 500 m for continuity) should be added to the figure(s) (especially Fig 1) as well.

L62: is the choice of 100 m separation for swath mode, vs 400 m for POCA mode purely arbitrary?

L74 "This is not an issue…" What is not an issue?

L80: Stray sentence. And I think that the comma should be a semicolon.

L84-85: "Data from both radar … have been used." I take it that this means that these data were used in the MEaSUREs velocity assessments?

L105-108: it might be good to see the velocity field somewhere. It would help with this sentence and to help the reader see the choices you made for the sides of your glacier domain.

L108: "The average ice thicknesses … are … speeds … are … etc. Also, be consistent with Oxford commas throughout the paper.

L109: Mankoff et al., 2018 (and Mankoff et al 2019, cited in the Intro) are missing from refs

L112: again, the comma here should be a semicolon. While this phrasing works conversationally, it's really just 2 clauses connected poorly.

L123: I'd pull this equation out of the paragraph.

L123-136: the subscripts disappear for rho or lose an underscore associated with ice_in; check for consistency. For rho_w in equation 1 and in the text in L123: should this be included to make the units cancel?

L182-183: The basin 2 and basin 3 run-off values look comparable for the entire time series (e.g., 2013-2014). It is fair to say, however, that the error circles are converging; perhaps discuss that? I understand that the larger basin runoff must include the smaller basin runoff, and the smaller basin will never outpace the larger basin, but why do the errors converge?

L193: Overall, the error discussion is pretty thorough. More papers should have this level of discussion on errors.

L195-196 and L205-207: why is accumulation included in L195 (and not firn densification), while firn densification is included in L206 (and not accumulation)? I am sure that I am missing something, but perhaps some language here to help the reader would be good.

L207-208: I don't understand this sentence; some description of this in this reference (as opposed to just in Gray et al., 2019) is needed.

L249-252: some mention should be made to 1) CryoSat track spacing (it's denser up north) and 2) the broad nature and orientation (relative to satellite ground tracks) of Humboldt. Both of these things contribute to more satellite data volume.

L269: 'ICESat-2'

L279: Here you use what I have always considered the more common way of referring to 'CryoSat-2'. And BedMachine now has a capitalized 'M'. Which is it? Perhaps be consistent here.

Figure 1: as mentioned above, it would be good to have some sort of grid. Also helpful would be to add a contours (like in Figure 3) and include an additional contour between 1000 m and the water.

Figure 2a: remind people of what the 'added values' are, or where they come from, or just point them to the statement in the text.
Figure 2c: for consistency, I'd add 'from 3 basins from title'.
Figure 2d: my image has 2 x-axes.
Figure 2e: did you exclude basin 3 because basin 2 and 3 have such similar results in 2d?
Caption: "…the potentially uncertainty…"

Figure 3a: add contour down low

---

## Referee Comment (RC2) · Anonymous Referee #2 · 12 Nov 2020

This manuscript presents a concise and well-thought-out analysis of bass balance for a small but interesting piece of Humboldt Glacier. Although it does not try to solve the entire problem to ice-sheet mass balance, it contains a thorough and detailed analysis that would be a good model for other studies of the same type. I was happy with the way the material was presented, except for a piece of the analysis of the altimetry that I didn't understand, and that I think needs either more detail or a more convincing argument.

The section that I found problematic was the treatment of the mass balance for the high-elevation portion of the basin. Here, Dr. Gray writes: "Thirty-day height change data

derived from all the CryoSat POCA data for basin 3 above 1300 m (Fig. 3) shows that there was a melt season height decrease in this area only in 2012, 2015 and 2019 (Fig. 3B). As there is no evidence of run-off to lower elevations or of an associated change in speed, I ascribe these height changes predominantly to surface melt and sub-surface refreezing, i.e., firn densification, and reduce the volume loss using these values so that the mass loss can be estimated. The average summer height losses associated with densification (Fig. 3B) were 0.42 ±0.08 m (2012), 0.45 ±0.08 m (2015) and 0.4 ±0.08 m (2019) based on the 44,756 POCA height results spanning the nine years and the area of basin 3 above 1300 m."

It seems to me that we should expect to see surface lowering during the summer. Even in an area of a glacier where the velocity is constant, for steady state the accumulation need only balance ice flow in the annual average. During the winter, we would expect to see accumulation in excess of that needed to balance ice flow and consequent surface rise, and during the summer, we would expect to see too little accumulation to balance ice flow, and consequent surface lowering. It seems that to assess the contributions of accumulation and densification to volume change we need external information. It may be that the MAR data provide clues about this, and there's been some extra analysis that didn't make it into the manuscript, but I didn't see a strong argument for why all the surface lowering should be due to firn densification. Absent this argument, might it make more sense to remove the area upstream of 1300 m from the analysis? It seems that it's not the source of significant runoff.

My other comments are editorial.

Line 18: Should give an example of the use of the model, not just a reference to the model.

Line 20: "height when" -> "height with radar altimeters when"

Line 37: small -> weak

Line 38: "to the supra-glacial" -> "for the supra-glacial"

Line 38: add "of the lake" after "estimation"

Line 39: "By using" -> By selecting

Line 43: This paragraph needs an introductory sentence that summarizes the method and helps provide some context for the trade off between patch size and accuracy.

Line 43: "data is" -> "data are"

Line 47: rather than saying that the size of the window can be increased, say that you increased it (otherwise it's ambiguous what was done in the study).

Line 51: add comma after "this period"

Line 52: add comma after "369 days"

Line 53: add comma between "next" and "many

Line 55: "year-to-year work" -> "year-to-year differences"

Line 57: "has provided" -> "provides

Line 58: add comma after moderate

Line 61: "samping is 2.4 km" -> "bin spacing is increased to 2.4 km.

Line 66: "good weather data" -> "accurate weather data"

Line 74: change-> vary

Line 103: thickness and speed should both be plural.

143-144: "are available as mm" -> "are provided in units of mm"

144: I think the units are most likely mm water equivalent (not per square meter). This is numerically the same as kg / m^2.

145: "at 150 m" -> "on a 150-m grid"

164: "is less" -> "are fewer

165: "less than that" -> "smaller than those"

166: add comma after "four gates"

173: comma after "balance"

208: delete quotes around 30-day

235: issue->question

Data availability: Need to provide a source for the MAR accumulation data.

---

## Author Comment (AC1) · 1 Dec 2020

I would like to thank both reviewers for the time and care they have taken with their reviews. I think responding to their comments and criticisms has improved the paper. In the following the reviewer comments are in italics and the responses are in standard font.

Reviewer 1.

*This paper uses CryoSat-2radar altimetry (and OIB ice thickness data and MEaSUREs velocity data) to determine glacier runoff in a northern Greenland glacier basin. The Humboldt is used as a test case for this method.*

*One general comment that I have is that there needs to be more of a discussion on where this method may be applicable. The author describes the melt/water environment as being the key limitation. I'd suggest that here are other limitations that include: latitude, trend and width of the glacier valley, and OIB data availability.*

This is fair comment and section 4.3 'Applicability of this method' has been expanded to cover the other limitations. The revised text is shown below.

**4.3 Applicability of this method**

Although data on ice thickness, surface height and velocity are now readily available on the internet, this method for the estimation of water run-off associated with a large glacier does have some limitations; the downstream gate close to the glacier terminus needs to be relatively large (~ 10+ km), and ice thickness and velocity data are required for all the gates. While ice thickness close to the terminus has been measured for many of the large Greenland outlet glaciers, often the ice thickness upstream has not been estimated as accurately. As described earlier, obtaining height change data close to the terminus of many of Greenland's larger glaciers can be challenging due to the relatively large slopes and surface roughness. The derivation of the height change at the gates improves with the number of height estimates. Consequently, the error associated with gate height change may increase in southern Greenland as the coverage by CryoSat degrades due to the sub-satellite track divergence.

 If there was a steady input of englacial or subglacial water at the upper gate and an equal amount of water leaving at the lower gate at the same time this would not be detected with this method. However, around Greenland there is both ice discharge and water run-off, and, as the run-off is predominantly seasonal, it can be estimated with this method. However, the method will be more challenging for those areas and glaciers with strongly divergent upstream flowlines.

*While this is intended to be a brief communication, the Height change estimation using CryoSat section probably needs more information to have this paper stand a little more on its own.*

Some extra information has been added particularly in relation to the method by which the temporal height change is estimated. The improved section is added below:

**2.1 Height change estimation using CryoSat**

[revised manuscript text omitted]

*The paper is fairly well written, although it needs a bit of polish that usually comes from having multiple authors and multiple sets of eyes on a manuscript.*

Further editing has been done and the current paper has been read by a person familiar with paper editing.

*I don't think that the title should have a period*

The period has been removed.

*L19: "...synthetic aperture radar (SAR), interferometric SAR (SARIn)..."*

The CryoSat SARIn mode uses what is called 'unfocussed synthetic aperture processing' but to avoid unnecessary jargon and still be accurate in the description of the system the text has been expanded as below…

The interferometric mode on the European Space Agency (ESA) CryoSat satellite was developed in part to alleviate the problems associated with measuring glacial ice height with radar altimeters when surface slopes are relatively large, e.g., at the periphery of the Greenland Ice Sheet. Coherent processing is used in the 'SARIn' mode to achieve an along-track resolution of ~ 380 m, and two cross-track antennas are used for footprint geocoding using interferometric processing (Parrinello et al., 2018).

*L30: I am concerned about differences between baseline C and D. Did you see this? How would differences affect these results?*

The main difference between baselines -C and -D L1b files is the file format, the only important change which could affect the swath processing output is the small

improvement in the satellite roll angle provided in baseline-D. However, prior to the introduction of the baseline-D L1b product, ESA had provided the revised roll angles in an ftp site. Whenever baseline-C files were used in this work the revised roll angles were also used. The change from baseline-C to baseline-D has essentially no impact on the 'POCA' results. The revised text is…

CryoSat baseline-C and -D intermediate level SARIn (L1b) data collected from the summer of 2010 to the end of 2019 have been used in this work. Baseline-D files contain small improvements to the satellite roll angle which could affect the output of the swath processing algorithm. However, ESA provided corrections prior to the introduction of baseline-D which were used in the current work. Processing to terrain height and height change was done using the methods described in Gray et al., 2015, 2019.

*L36-38: This sentence is a bit long and hence a bit confusing. Needs an edit.*

The text for this part of the paper is now as follows…

With swath mode processing it is important to select conditions which will minimize the contribution from any unwanted range ambiguous region. For example, when the height and height change of the surface of supra-glacial lakes were mapped using CryoSat swath mode data (Gray et al., 2017), the relatively flat lake provided a strong reflecting surface so that the returns from the lake dominated over any range-ambiguous regions. Consequently, the differential phase reflected the cross-track look angle for the supra-glacial lake and allowed accurate geocoding and height estimation of the lake.

*L43-50: Somewhere in here, it might be good to acknowledge that this outlet glacier is at a high latitude(79.5 N), providing more data points than you would get from the southern part of the island, simply based on converging satellite ground tracks. It might be good to add a grid or some indication of this (latitude/longitude) in your figure(s).Further, this is a large glacier, with a wide terminus, providing more data points than you would get from more constrained outlets.*

Again, this is a fair criticism and I have expanded section 4.3 'Applicability of the method' to cover this and other issues related to when this approach to estimating run-off is possible.

Both Figure 1 and 3A have been modified to provide height contours, velocity vectors and a latitude/longitude grid has been plotted on the expanded insert map of Greenland in figure 1.

*L49: Why was this time range selected? It seems a little later (both start and finish) than I expected. I now see (on L242) why the October date is chosen; what about the start?*

The time period for the year-to-year CryoSat height change (July 1 to Oct. 15) is somewhat arbitrary but was selected such that there was adequate data during the summer-fall period for the year-to-year comparison. Some tests were done to see how

the results might change as the start and stop dates were changed. This showed that the results were relatively insensitive to the precise start-stop dates.

*L60: Where is the 600 m contour? Below 1000 m the contours can bunch up quickly. This contour (or perhaps the 500 m for continuity) should be added to the figure(s) (especially Fig 1) as well.*

Both figures 1 and 3A have been changed to include the 500 m contour.

*L62: is the choice of 100 m separation for swath mode, vs 400 m for POCA mode purely arbitrary?*

Not really. The choice is based on the footprint size, the 'density' (number per $km^2$) of the centres of the footprints, and the time required to run the search programme. Again, some years ago I experimented with different separation criteria and found that the results were relatively insensitive to this number. This is covered in earlier papers and to keep the length of the communication under control I have not explicitly addressed this concern in the rewrite.

*L74 "This is not an issue..." What is not an issue?*

This referred to the need to use height results over a relatively large area to get good statistics for the results for the average 30-day height variation. The 'large area' would be an issue if the surface conditions and height change were changing significantly with position (as it does close to the terminus). I do not expect that the average height change changes much with position in the accumulation zone, hence we can use a relatively large area to study it. The second reviewer also had a more general concern with this section which has led to a more extensive rewrite which also addresses this question. The specific part of the rewrite is now…

To search for height change associated with firn densification it is necessary to use the 30-day temporal sampling and a relatively large area in the accumulation zone, > ~$10^3$ $km^2$. Using a relatively large area is not an issue as conditions in the accumulation zone vary relatively slowly with position.

*L80: Stray sentence. And I think that the comma should be a semicolon.*

The corrected sentence has been moved to the results section.

*L84-85: "Data from both radar...have been used." I take it that this means that these data were used in the MEaSUREs velocity assessments?*

Yes, the data in the NSIDC MEaSUREs Greenland ice velocity web site has been derived from both radar and optical satellites.

*L105-108: it might be good to see the velocity field somewhere. It would help with this sentence and to help the reader see the choices you made for the sides of your glacier domain.*

Figure 3 now includes a 'quiver' plot of the velocity vectors in which the length of the vector is proportional to the square root of the speed.

*L108: "The average ice thicknesses... are...speeds ... are... etc. Also, be consistent with Oxford commas throughout the paper.*

The wording here has been changed to the following and care has been taken throughout the paper to improve my use of commas…

When calculating the flux, the surface speed is normally used as the depth averaged speed whenever the surface speed is ~ 100 m/year or larger (Mankoff et al., 2020). The flux estimates through gates 3 and 4 have used fractions of the surface speed to account for the possibility that the depth averaged speed is less than the surface speed. For gate 3 the fractions were 0.95, .975, and 1, and 0.9, .95 and 1 for gate 4. The different fractions are used to estimate the potential error arising from the uncertainty in the depth averaged velocity.

*L109: Mankoff et al., 2018(and Mankoff et al 2019, cited in the Intro) are missing from refs*

The up-to-date reference has been added.

*L112: again, the comma here should be a semicolon. While this phrasing works conversationally, it's really just 2 clauses connected poorly.*

The wording has been changed as above.

*L123: I'd pull this equation out of the paragraph.*

Done.

*L123-136: the subscripts disappear for rho or lose an underscore associated with ice_in; check for consistency. For rho_win equation 1 and in the text in L123: should this be included to make the units cancel?*

Yes, this omission was careless and has been fixed.

*L182-183: The basin 2 and basin 3 run-off values look comparable for the entire time series (e.g., 2013-2014). It is fair to say, however, that the error circles are converging; perhaps discuss that? I understand that the larger basin runoff must include the smaller basin runoff, and the smaller basin will never outpace the larger basin, but why do the errors converge?*

The wording here has been changed and expanded here primarily to address a concern from reviewer 2. However, this point is addressed specifically in the errors section as follows…

It is important to note the difference in the way that errors propagate for the different components of the equation for run-off. The mass change term is based on the altimetry derived volume change and the error in the comparing the volume between any two years is approximately the same, i.e., the error in the volume or mass change between 2011 and 2012 is essentially the same as between 2011 and 2019. But this is not the case for the flux estimates where an error in, e.g., the cross-sectional area of an ice gate, will accumulate with time such that the error in flux over nine years could be nine times the error over one year. This is the source of the increasing error estimates with time in Fig. 2D.

*L193: Overall, the error discussion is pretty thorough. More papers should have this level of discussion on errors.*

*L195-196 and L205-207: why is accumulation included in L195 (and not firn densification), while firn densification is included in L206 (and not accumulation)? I am sure that I am missing something, but perhaps some language here to help the reader would be good.*

The uncertainty related to firn densification is implicit in the error assigned to the mass change term. The error in mass change has two components; the error in volume change and the contribution related to the uncertainty in density associated with the volume change. The wording close to lines 196 and 205 has been modified and clarified to better explain this. The new version has been expanded also to address a major concern from reviewer 2.

*L207-208: I don't understand this sentence; some description of this in this reference (as opposed to just in Gray et al., 2019) is needed.*

This sentence has been removed. The rewritten section on processing the CryoSat data now includes an extra paragraph describing the way that temporal height change is calculated and the estimation of the random error in the results.

*L249-252: some mention should be made to 1) CryoSat track spacing (it's denser up north) and 2) the broad nature and orientation (relative to satellite ground tracks) of Humboldt. Both of these things contribute to more satellite data volume.*

This has been addressed in the expanded section 'Applicability of this method'.

*L269: 'ICESat-2'*

Fixed.

*L279: Here you use what I have always considered the more common way of referring to 'CryoSat-2'. And BedMachine now has a capitalized 'M'. Which is it? Perhaps be consistent here.*

The inconsistencies have been fixed. For simplicity I have omitted the -2 in referring to CryoSat.

*Figure 1: as mentioned above, it would be good to have some sort of grid. Also helpful would be to add a contours (like in Figure 3) and include an additional contour between 1000 m and the water.*

Figure 1 and 3A have been changed as requested.

*Figure 2a: remind people of what the 'added values' are, or where they come from, or just point them to the statement in the text.*
*Figure 2c: for consistency, I'd add 'from 3 basins from title'.*
*Figure 2d: my image has 2 x-axes.*

These changes and additions to Fig. 2 have been done.

*Figure 2e: did you exclude basin 3 because basin 2 and 3 have such similar results in 2d?*

Yes.

*Caption:"...the potentially uncertainty*

The caption has been changed.

*Figure3a: add contour down low*

500 m contour added.

---

## Author Comment (AC2) · 1 Dec 2020

I would like to thank both reviewers for the time and care they have taken with their reviews. I think responding to their comments and criticisms has improved the paper. In the following the reviewer comments are in italics and the responses are in standard font.

Reviewer 2.

*This manuscript presents a concise and well-thought-out analysis of bass balance for a small but interesting piece of Humboldt Glacier. Although it does not try to solve the entire problem to ice-sheet mass balance, it contains a thorough and detailed analysis that would be a good model for other studies of the same type. I was happy with the way the material was presented, except for a piece of the analysis of the altimetry that I didn't understand, and that I think needs either more detail or a more convincing argument.*

*The section that I found problematic was the treatment of the mass balance for the high-elevation portion of the basin. Here, Dr. Gray writes: "Thirty-day height change data derived from all the CryoSat POCA data for basin 3 above 1300 m (Fig. 3) shows that there was a melt season height decrease in this area only in 2012, 2015 and 2019 (Fig.3B). As there is no evidence of run-off to lower elevations or of an associated change in speed, I ascribe these height changes predominantly to surface melt and sub-surface refreezing, i.e., firn densification, and reduce the volume loss using these values so that the mass loss can be estimated. The average summer height losses associated with densification (Fig. 3B) were 0.42±0.08 m (2012), 0.45±0.08 m (2015) and 0.4±0.08 m (2019) based on the 44,756 P OCA height results spanning the nine years and the area of basin 3 above 1300 m."*

*It seems to me that we should expect to see surface lowering during the summer. Even in an area of a glacier where the velocity is constant, for steady state the accumulation need only balance ice flow in the annual average. During the winter, we would expect to see accumulation in excess of that needed to balance ice flow and consequent surface rise, and during the summer, we would expect to see too little accumulation to balance ice flow, and consequent surface lowering. It seems that to assess the contributions of accumulation and densification to volume change we need external information. It maybe that the MAR data provide clues about this, and there's been some extra analysis that didn't make it into the manuscript, but I didn't see a strong argument for why all the surface lowering should be due to firn densification. Absent this argument, might it make more sense to remove the area upstream of 1300 m from the analysis? It seems that it's not the source of significant runoff.*

I acknowledge that summer surface lowering in the accumulation zone need not be due solely to firn densification. Also, as the results show that the run-off originating from the area above 1300 m is minimal, the issue of the source of the summer height decrease in the accumulation zone in some years is not central to the contribution of the paper, and could be left out. Nevertheless, I would still like to include a rewritten section on the observed height change in the accumulation zone, and the possibility that in some years this could be indicative of firn densification. The expanded rewrite is now a little more speculative, but I hope still persuasive and useful to readers. Data from satellite altimeters, both optical and radar, represent a very direct way to track ice cap and ice sheet volume change but there is an issue in converting this to the more important mass change. Most papers using firn densification models to do this, e.g.

for radar altimetry; McMillan, M., et al. (2016), Geophys. Res. Lett., 43, 7002–7010. 2016, and for laser altimetry; Smith et al., Science, June 2020, Vol. 368, Issue 6496, pp. 1239-1242. But the results can only be as good as the reanalysis of the input weather data, which are very sparce for the large ice sheets. I think any approach which can contribute to this problem should be explored, especially a direct method using satellite data. Here the sequence of SAR imagery spanning the summer of 2019 (supporting material) shows conclusively that there was surface snow melt into the accumulation zone of the test area in this year. The record of yearly melt conditions over Greenland on the NSIDC web site also documents the unusual conditions in this area in the summers of 2012, 2015 and 2019. The unusually warm conditions for 2012 are well known. For summer 2015; (from http://nsidc.org/greenland-today/2015/11/), '….a surge in melting in late June and all of July as very warm conditions prevailed along the far northern and northwestern coast,…'. And for 2019, (from http://nsidc.org/greenland-today/2019/11/ ) ; '…(melting) was particularly intense along the northern edge of the ice sheet, where compared to the 1981 to 2010 average, melting occurred for an additional 35 days.'

The high temporal resolution (30-day) height change data for the 44,756 points in the upper accumulation part of the test area (Fig.3B) shows an overall height decrease from the fall of 2010 to the fall of 2019 of ~ 1.5 m with numerous small rises and dips. However, there are three relatively large height decreases of ~0.5 m, now marked with red arrows in the revised Fig.3B, which correspond to the melt seasons in 2012, 2015 and 2019. The height decreases could be associated with a relatively sudden change in ice speed but no such summer spike in speed has been observed at these elevations. Consequently, I think we can confidently associate these anomalous height decreases to the unusually warm summers in this area for these years. However, if surface melt did lead to the height decrease where did the water go? Again, I think the high-resolution SAR imagery helps show that there are none of the clues that one would normally associate with run-off to a lower elevation, e.g. surface streams, moulins or supra-glacial lakes. The suspicion then is that the water percolated downwards and ultimately became refrozen, thus leading to surface lowering and increased firn density of the near surface layer. If we assume that the height change reflects firn densification through surface melt and percolation, then calculating the volume change to mass change is now straightforward: The summer volume loss in this area does not represent a mass loss, and the summer dips in elevation in 2012, 2015 and 2019 can be discounted in calculating the mass losses.

The rewritten part of the text in the results section is added below for convenience…

The cumulative net run-off for the three basins (Fig. 2D) is estimated based on the ice flux difference between input and output, the accumulation and the net change in basin mass, as described in section 2.4 above. By the Fall of 2019, the cumulative run-off for basin 2 is comparable to that for the larger basin 3. As the larger basin contains the smaller one, the run-off from the larger basin cannot be less than the smaller one implying that most of the run-off originates from below gate 3 in all years. However, when converting the yearly volume change to mass change in the accumulation zone care should be taken to account for changing summer weather conditions and the impact this may have on firn compaction and therefore, near surface density.

Firn densification models can be used to improve the volume to mass change estimation, e.g. McMillan, et al. (2016) and Smith et al., (2020), but the results can only be as good as the reanalysis of the input

weather data, which are very sparce for the large ice sheets. Here, a straightforward correction has been carried for three years when anomalous height decreases were observed for the summers of 2012, 2015 and 2019. Figure 3 shows the positions of 44,756 height estimates above 1300 m in basin 3, and Fig. 3B shows the average height change sampled at 30-day intervals from the Fall of 2010 to the Fall of 2019. The three red arrows indicate the anomalous height decreases in the summers of 2012, 2015 and 2019. In an idealized situation, the surface height would not change for an ice sheet in equilibrium, and the slow snow accumulation would be balanced by the slow downslope movement of the ice. However, the detected height change data can be affected by temporal changes in accumulation, downslope ice speed and near surface conditions including summer firn densification. A sequence of Sentinel SAR imagery spanning the summer of 2019 (see the supporting material) shows that there was surface snow melt extending up into the accumulation zone of the test area in this year. The NSIDC 'Greenland Ice Sheet Today' web site documents the melt conditions over Greenland and the unusual conditions in this area in the summers of 2012, 2015 and 2019. The unusually warm conditions for 2012 are well known. For the summer 2015; (from http://nsidc.org/greenland-today/2015/11/), '....a surge in melting in late June and all of July as very warm conditions prevailed along the far northern and north-western coast,...'. And for 2019, from http://nsidc.org/greenland-today/2019/11/ ; '...(melting) was particularly intense along the northern edge of the ice sheet, where compared to the 1981 to 2010 average, melting occurred for an additional 35 days'. Consequently, the anomalous height decreases in this area can be linked to the unusually warm summers in 2012, 2015 and 2019. While the height decreases could be due to a relatively sudden change in downstream ice speed no such summer spike in speed has been observed at these elevations. As there are none of the clues that one would normally associate with run-off to a lower elevation, e.g. surface streams or supra-glacial lakes, the most likely explanation for these three summer height decreases is surface melting, percolation of the melt water and subsurface refreezing. When calculating the volume change to mass change I assume, therefore, that the height losses of 0.42 ±0.08 m (2012), 0.45 ±0.08 m (2015) and 0.4 ±0.08 m (2019) were due to firn densification and I correct the yearly volume change accordingly. The error associated with this assumption is hard to evaluate but an additional error of ±10 cm has been included to account for unknown biases in the height data (section 3.2 below).

*My other comments are editorial.*

*Line 18: Should give an example of the use of the model, not just a reference to the model.*

The Fettweis 2017 reference does show how the model is used… ('Reconstructions of the 1900–2015 Greenland ice sheet surface mass balance using the regional climate MAR model')

*Line 20: "height when" -> "height with radar altimeters when"*

Phrase added.

*Line 37: small -> weak*

This part was rewritten…

For example, when the height and height change of the surface of supra-glacial lakes were mapped using CryoSat swath mode data (Gray et al., 2017), the relatively flat lake provided a strong reflecting surface so that the returns from the lake dominated over any range-ambiguous regions. Consequently,

the differential phase reflected the cross-track look angle for the supra-glacial lake and allowed accurate geocoding and height estimation of the lake.

*Line 38: "to the supra-glacial" -> "for the supra-glacial"*

Changed, as above

*Line 38: add "of the lake" after "estimation"*

Done.

*Line 39: "By using" -> By selecting*

Changed.

*Line 43: This paragraph needs an introductory sentence that summarizes the method and helps provide some context for the trade off between patch size and accuracy*

An introductory sentence has been added…

In calculating ice height and temporal height change we need to be able to change both the area over which the change will be measured and also the time interval between average height estimates. There is a trade-off…

*Line 43: "data is" -> "data are"*

Fixed.

*Line 47: rather than saying that the size of the window can be increased, say that you increased it (otherwise it's ambiguous what was done in the study).*

Fixed.

*Line 51: add comma after "this period"*

Done.

*Line 52: add comma after "369 days"*

Done.

*Line 53: add comma between "next" and "many*

Done.

*Line 55: "year-to-year work" -> "year-to-year differences"*

Changed.

*Line 57: "has provided" -> "provides*

Changed.

*Line 58: add comma after moderate*

Comma added.

*Line 61: "samping is 2.4 km" -> "bin spacing is increased to 2. .4 km.*

Changed.

*Line 66: "good weather data" -> "accurate weather data"*

Changed.

*Line 74: change-> vary*

Changed.

*Line 103: thickness and speed should both be plural.*

Corrected.

*143-144: "are available as mm" -> "are provided in units of mm"*

Yes, corrected.

*144: I think the units are most likely mm water equivalent (not per square meter). This is numerically the same as kg / mˆ2.*

As above

*145: "at 150 m" -> "on a 150-m grid"*

Changed.

*164: "is less" -> "are fewer*

*165: "less than that" -> "smaller than those"*

164 and 165 Fixed, as suggested.

*166: add comma after "four gates"*

Done,

*173: comma after "balance"*

Done.

*208: delete quotes around 30-day*

That sentence has gone in the rewrite.

*235: issue->question*

*Data availability: Need to provide a source for the MAR accumulation data.*

The link to the ftp site has been added to the data availability section.

---

## Referee Report (RR1)

This paper uses CryoSat-2 radar altimetry (and OIB ice thickness data and MEaSUREs velocity data) to determine glacier runoff in a northern Greenland glacier basin. The Humboldt is used as a test case for this method.

The author has directly and adequately addressed many of the points that I raised in my previous review. The changes based on these comments and those of the second reviewer have greatly improved the manuscript.

Here are a couple minor comments:

Eq 1 should probably be followed by a comma

Eq 2 has a stray '.' (between the 'A' and the rho_w), which needs to move to the end of the equation.